Global stock structure of the Silky shark (Carcharhinus falciformis, Carcharhinidae) assessed with high-throughput DNA sequencing

Kraft Derek W. Kraftd@hawaii.edu 1
Conklin Emily E. 1
Freel Evan B. 1
Hutchinson Melanie 1
Spaet Julia L.Y. 2
Toonen Robert J. 1
Forsman Zac H. 1 3
Grant Michael I. 4
Filmalter John David 5
Hyde John R. 6
Gulak Simon J.B. 7
Bowen Brian W. 1
1 University of Hawaiʻi, Hawaiʻi Institute of Marine Biology , Kãne‘ohe , Hawaiʻi , USA
2 Department of Zoology, University of Cambridge, Evolutionary Ecology Group , Cambridge , United Kingdom
3 King Abdullah University of Science and Technology (KAUST), Reefscape Restoration Initiative , Thuwal , Kingdom of Saudi Arabia
4 James Cook University, Centre for Sustainable Tropical Fisheries and Aquaculture and College of Science and Engineering , Townsville , Queensland , Australia
5 South African Institute for Aquatic Biodiversity , Makhanda , South Africa
6 National Marine Fisheries Service, National Oceanic and Atmospheric Administration , La Jolla , CA , USA
7 Riverside Technology, Inc. for NOAA Fisheries , Fort Collins , CO , United States of America
Banaszak Anastazia
Electronic publication date: 2025 Jul 7
Publication date: 2025
Volume: 13
Electronic Location ID: e19493
Received 2023 Aug 31; Accepted 2025 Apr 28
Copyright: ©2025 Kraft et al.
Copyright year: 2025
Copyright holder: Kraft et al.
License: This is an open access article distributed under the terms of the Creative Commons Attribution License, which permits unrestricted use, distribution, reproduction and adaptation in any medium and for any purpose provided that it is properly attributed. For attribution, the original author(s), title, publication source (PeerJ) and either DOI or URL of the article must be cited.
License URL: https://creativecommons.org/licenses/by/4.0/

Keywords: Marine fisheries, Elasmobranch, Marine fishes, Phylogeography, Pool-seq, Stock structure, Bycatch, Shark fisheries, Fisheries management, Shark genetics

Funding: University of Hawaiʻi Sea Grant College Program NA14OAR4170071 NOAA Office of Sea Grant, Department of Commerce Save Our Seas Foundation U.S. National Science Foundation OCE-1558852, OCE-1924604 International Seafood Sustainability Foundation UNIHI-SEAGRANT-4861 from the University of Hawaiʻi Sea Grant Program This paper is funded by a cooperative agreement with the National Oceanic and Atmospheric Administration, Project R/SS-19PD, which is sponsored by the University of Hawaiʻi Sea Grant College Program under Institutional Grant No. NA14OAR4170071 (Brian W. Bowen) from NOAA Office of Sea Grant, Department of Commerce. Additional funding was provided by the Save Our Seas Foundation (Derek W. Kraft), and the U.S. National Science Foundation (http://www.nsf.gov/) grant OCE-1558852 (Brian W. Bowen) and OCE-1924604 (Robert J. Toonen). The International Seafood Sustainability Foundation funded the sample collection by Melanie Hutchinson and John David Filmalter in the IDO, SCP and AFR during chartered research cruises aboard commercial fishing vessels. This was also supported by the UNIHI-SEAGRANT-4861 from the University of Hawaiʻi Sea Grant Program. The views expressed herein are those of the authors and do not necessarily reflect the views of NOAA or any of its subagencies. The funders had no role in study design, data collection and analysis, decision to publish, or preparation of the manuscript.

==============================
Silky shark (Carcharhinus falciformis, Carcharhinidae) numbers have declined steeply in recent decades due to the fin fishery and bycatch in pelagic fisheries. Due to a lack of data on stock delineations, this species is currently managed in ocean-spanning jurisdictions defined by regional fisheries management organizations (RFMOs). Here we investigate the global stock structure of silky sharks and compare population structure to the four RFMO boundaries. Using high-throughput sequencing from pooled individuals (pool-seq) based on 628 specimens collected opportunistically across 11 circumglobal regions, yielding 854 nuclear single nucleotide polymorphisms (SNPs) and 23 mtDNA SNPs. Results indicate significant population genetic structure between all 11 regional sampling locations, with discriminant analysis of principal components (DAPC) identifying seven discrete groups. Within the Atlantic and Indo-Pacific Oceans, FST values ranged from 0.014 to 0.035 for nuclear (nDNA) markers, and from 0.012 to 0.160 for whole mtDNA genomes, with much higher values between than within oceans (mtDNA: 0.383–0.844, nDNA: 0.042–0.078). Using an analysis of molecular variance (AMOVA) framework, 22.24% of the observed population variance is explained by RFMOs, 32.1% is explained among ocean basins, and 34.81% is explained by the DAPC-identified groups. We find significant population genetic structure within the jurisdiction of every RFMO, from which we have more than a single sampling site. Our genomic-scale results indicate discordance between population genetic structure and RFMOs, highlighting the need for a detailed study to accurately identify stock boundaries.

Introduction

Sharks occupy a variety of marine habitats and fill many predatory roles in coastal and oceanic habitats, yet they are declining at an alarming rate (Dulvy et al., 2021; Kottilli et al., 2023; Dedman et al., 2024). Among the largest declines, populations of pelagic sharks have declined by ∼70% since the 1970’s, due to historic targeted fisheries for shark, and ongoing bycatch in tuna and billfish fisheries (Pacoureau et al., 2021). The silky shark (Carcharhinus falciformis, Carcharhinidae), is a semi-pelagic species inhabiting tropical and sub-tropical waters, where it overlaps with intensively targeted tuna stocks around the globe. In addition to being targeted for fins and meat in some fisheries, silky sharks are often captured as bycatch in tuna fisheries, where for example, in the western and central Pacific Ocean tropical tuna purse seine fisheries can account for >90% of the shark bycatch in fish aggregating device (FAD) associated sets (Lawson, 2011). This species is also one of the most abundant species in shark fin markets (Bonfil, Mena & De-Anda, 1993; Clarke et al., 2006; Cardeñosa et al., 2018). Their ‘slow’ life history characteristics (low fecundity, late maturity, and long lifespans) make silky sharks susceptible to overharvesting (Grant et al., 2019). Owing to fishery induced declines across ocean basins (Clarke et al., 2018; Urbina et al. 2018; Neubauer et al., 2024) it is currently assessed as vulnerable to extinction on the International Union for Conservation of Nature (IUCN) Red List of Threatened Species (Rigby et al., 2021). Highlighting the need for coordinated international management efforts for this species, the silky shark is listed on Appendix II of the Convention on International Trade in Endangered Species of Wild Fauna and Flora (CITES) and Appendix II of the Convention on the Conservation of Migratory Species of Wild Animals (CMS). Without conservation management interventions, this species may continue to decline and could follow the now critically endangered oceanic whitetip shark (Carcharhinus longimanus, Carcharhinidae) which has a similar ecology, range, and exploitation history (Rigby et al., 2019).

Due to the pelagic nature of silky sharks and interaction rates with tuna fisheries, their management comes primarily from four tuna-oriented Regional Fisheries Management Organizations (RFMOs) that manage and conserve highly migratory tuna and associated species. The Indian Ocean Tuna Commission (IOTC), Western Central Pacific Fisheries Commission (WCPFC), Inter-American Tropical Tuna Commission (IATTC), and International Commission for the Conservation of Tuna (ICCAT), each manage silky sharks as a single stock within their jurisdictions (Fig. 1). The assumption of a single stock within political boundaries implies that heavy fishing in one part of the RFMO jurisdiction will be replenished by sharks from other areas within the RFMO jurisdiction. Since RFMO jurisdictional areas span ocean basins, there may be distinct stocks within an RFMO. If areas where heavy fishing occurs rely on self-recruitment, then vulnerability to overfishing may not be evenly distributed. This stock structure would have alarming implications regarding population integrity and management approach for stock recovery, as smaller stocks are much more vulnerable to overfishing (Hutchings & Reynolds, 2004). Defining stocks for highly migratory species remains a difficult task; however, testing for genetic population structure can resolve dispersal boundaries and provide an appropriate foundation for defining stock boundaries (Carvalho & Hauser, 1994; Ward, 2000; Ablan, 2006; Ovenden et al., 2015).

Figure 1 (A) Carcharhinus falciformis photographed in Mexico, ©Martin Vranken (reproduced from FishBase (Froese & Pauly, 2022)). (B) Map showing Silky shark (Carcharhinus falciformis) opportunistic sampling locations.

The colored areas represent the Regional Fisheries Management Organizations (RFMO) jurisdictional boundaries. Regional sampling locations are labeled with site abbreviation and sample size (in parentheses). RFMO abbreviations: IOTC, Indian Ocean Tuna Commission; WCPFC, Western and Central Pacific Fisheries Commission; IATTC, Inter-American Tropical Tuna Commission; ICCAT, International Commission for the Conservation of Atlantic Tuna. Sample site abbreviation: RDS, Red Sea; IDO, Indian Ocean; TAI, Taiwan; PNG, Papua New Guinea; SCP, South Central Pacific; NCP, North Central Pacific; EPAC, Eastern Pacific; GOM, Gulf of Mexico; NWA, Northwest Atlantic; BRA, Brazil; AFR, Western Africa.

Previous studies have examined the regional population structure of silky sharks; one recent microsatellite study reported significant population structure in the Indian Ocean (Li et al., 2023). Three additional studies used the mitochondrial control region (mtCR) DNA to resolve population connectivity. Two of the three mtCR studies reported weak but significant population structure across the Indo-Pacific (Galván-Tirado et al., 2013; Clarke et al., 2015). One study found population structure within the Atlantic (Domingues et al., 2017) and two found strong population structure between the Atlantic and Indo-Pacific (Clarke et al., 2015; Domingues et al., 2017). For the past two decades, studies to resolve population structure of elasmobranchs have focused on a handful of mitochondrial and microsatellite loci (reviewed in Domingues et al., 2017). However, advances in DNA sequencing technology have made it possible to investigate the entire mitochondrial and nuclear genome.

High-throughput sequencing is a powerful tool for examining population structure across tens of thousands of single-nucleotide polymorphisms (SNPs) throughout the genome. As sequencing costs have decreased through time, SNPs have become the preferred method to establish population structure for commercially important marine species (Hess, Matala & Narum, 2011; Albaina et al., 2013; Diopere et al., 2017; Puncher et al., 2018). Although sequencing costs have decreased, even low-coverage sequencing of genomes from thousands of individuals remains cost-prohibitive for most research labs, especially for non-model organisms that lack reference genomes (Nielsen et al., 2018; Kurland et al., 2019; Kraft et al., 2020). Furthermore, for many research questions, whole genomes are not only cost-prohibitive, but also unnecessary, since individual genotypes are converted to population-level allele frequency estimates for many population genetics analyses (Hartl & Clark, 1997; Bowen et al., 2014; Schlötterer et al., 2014). Thus, pooling individual DNA specimens into combined libraries before sequencing (pool-seq) is an affordable method that allows for accurate large scale genetic analysis (Futschik & Schlötterer, 2010; Rellstab et al., 2013; Schlötterer et al., 2014; Mimee et al., 2015; Nielsen et al., 2018; Kraft et al., 2020; Nunez et al., 2021; Chen et al., 2022). Here, we used high-throughput sequencing of pooled individuals to examine the population structure of silky sharks between regions within RFMOs to quantify if current management regimes adequately reflect the biology of this pelagic species (Fig. 1). We survey both the mitochondrial and nuclear genomes of individuals, sampled opportunistically from eleven regions to provide the most extensive global silky shark population assessment to date. Results of this population genetic survey provide a scientific basis for resolving whether there may be multiple silky shark stocks within RFMOs. Throughout this manuscript we will refer to genetic stock structure as population structure unless stated otherwise.

Methods

Portions of this text were previously published as part of a doctoral dissertation (Kraft, 2020).

Sample collection

Samples consisting of fin clips or muscle sections were collected from 628 silky sharks from the Indian, Pacific, and Atlantic Ocean basins spanning 2007–2016. These samples were collected by scientists and fishery observers aboard commercial fishing vessels, collected at landing sites, and at fish markets based on local fisheries (Fig. 1, metadata provided in Document S1). Samples from the Gulf of Mexico and the Northwest Atlantic were the same individuals analyzed in the mtDNA study by Clarke et al. (2015). After collection, specimens were stored in 80% ethanol or saturated salt (NaCl) buffer. Samples were assigned to pooled groups based on the location of capture, with each pool consisting of 33 to 74 individuals (Fig. 1). Pooled groups are referred to by their region of capture for the remainder of this manuscript.

DNA sequencing

DNA was extracted using the Qiagen DNeasy Blood & Tissue kit (Qiagen, Mississauga, ON, Canada), following the manufacturer’s instructions except the elution step, which was performed with HPLC-grade water in an iterative three-elution process using first 35 ul, followed by 50 ul, and finally 100 ul, for a final elution volume of 185 ul. To ensure intact, high-quality DNA, extracts were inspected on a 2% agarose gel using the Gel Doc E-Z System (BIO RAD, Hercules, California, USA). An additional aliquot of extracted DNA was prepared for quantification using an AccuClear Ultra High Sensitivity dsDNA Quantitation Kit (Biotium, Fremont, CA, USA) and quantified on a SpectraMax M2 (Molecular Devices, Sunnyvale, CA, USA). Equal amounts of DNA (ng/µl) per individual were added to regional pools to minimize contribution bias. Pooled libraries contained a total of 2000 ng of DNA total. The ezRAD (Toonen et al., 2013) library preparation followed the ToBo Laboratory PCR-free protocol (Knapp et al., 2016), to maintain equimolar contributions of DNA from individuals across each library and minimize potential bias (Anderson, Skaug & Barshis, 2014). This library preparation used the restriction enzyme DPNII, and the KAPA HyperPlus PCR-free Kit for adapter ligation (KAPA Biosystems, Wilmington, MA, USA). Libraries were sequenced using Illumina MiSeq 300-bp paired-end runs with v3 reagent kits, performed by the Hawaiʻi Institute of Marine Biology EPSCoR Core sequencing facility.

Genetic analyses

Raw data (GenBank BioProject #PRJNA997384) and code (Supplemental Document 3) for these analyses are made available to ensure transparency and repeatability of these analyses. Following Kraft et al. (2020), sequence libraries were first examined with MultiQC v 1.2 (Ewels et al., 2016) to assess sequence quality scores, sequence length distributions, duplication levels, overrepresented sequences, and other artifacts. After quality checks, raw paired-end reads were trimmed using Trimmomatic v 0.33 (Bolger, Lohse & Usadel, 2014) and mapped to either the mitochondrial genome reference or assembled nuclear genomic reference (see below) using the Burrows-Wheeler Alignment (BWA) mem algorithm (Li & Durbin, 2009). SNPs were identified using Freebayes v 1.0.2 (Garrison & Marth, 2012, https://github.com/ekg/freebayes). These programs are wrapped in the dDocent bioinformatics pipeline (Puritz, Hollenbeck & Gold, 2014). To analyze the mitochondrial genome, a previously published silky shark mitochondrial genome was used as a reference (GenBank accession number KF801102, Galvan-Tirado et al., 2016).

The dDocent pipeline was also used to analyze the nuclear dataset. A de novo contig assembly was constructed and optimized following standard dDocent assembly protocols, including reference assembly optimization steps (http://ddocent.com/assembly/). Any contigs in the nuclear data set that aligned to the mitochondrial genome were removed, as was any contig with <30x mean coverage.

Due to differences between individual-library analyses and pooled-library analyses, SNP calling in Freebayes was optimized with the addition of the ‘pooled-continuous’ option and the minimum minor allele frequency was set to 0.05. SNPs were analyzed with the pool-seq specific bioinformatics pipeline assessPool (github.com/ToBoDev/assessPool). This pipeline uses VCFtools v 0.1.14 and vcflib to filter SNPs (Danecek et al., 2011), and Popoolation2 v 1.2.2 to compare allele frequencies between populations by calculating pairwise FST values, and finally relies on Fisher’s exact tests to determine locus-specific significance of pairwise comparisons (Kofler, Pandey & Schlötterer, 2011). AssessPool v1.0.0 (Freel et al., 2024) was then used to organize, summarize, and produce visualizations of the data using RStudio (RStudio Team, 2020).

An analysis of molecular variance (AMOVA) test was also conducted on nDNA SNPs using the amova() function in the R package ade4 (as implemented in poppr), which calculates AMOVA based on allele frequencies (Dray & Dufour, 2007; Kamvar, Tabima & Grünwald, 2014). As pooled data does not provide access to individual genotypes, we used the within = F flag to avoid processing our data as haplotypes. The SNP data was first converted to Genlight format using the package vcfR (Knaus & Grünwald, 2017). To test the components of variance for statistical significance, Monte-Carlo permutation tests were used as described in Excoffier, Smouse & Quattro (1992) and as implemented in the function randtest.amova() in ade4.

To further assess population structure among and between ocean basins, we conducted a principal components analysis (PCA) using nDNA SNPs. To perform a PCA on pool-seq data, we constructed a matrix of major allele frequencies for each variant site (Mimee et al., 2015) using the R package vcfR (Knaus & Grünwald, 2017). We then conducted a PCA using the function prcomp() in the base R stats package (v4.0.3) and visualized these data using the R package ggfortify (v0.4.15) (Horikoshi & Tang, 2018). A discriminant analysis of principal components (DAPC) was also performed with the nuclear SNP data, following the methods outlined in Suchocki et al. (2023) using the function dapc() from the R package adegenet (Jombart, 2008; Jombart, Devillard & Balloux, 2010). Briefly, two DAPCs were performed. The first used de novo groups generated by k-means clustering to determine the optimal number of genetic clusters (K). Bayesian information criterion (BIC) selection using the find.clusters() function in adegenet determined the optimal k-value. The membership of each cluster within a given RFMO was recorded and compared to the second analysis using a priori groups based on the RFMO jurisdictions. We used a-scores from the optim.a.score() function in adegenet to determine the optimal number of principal components to retain in both DAPC analyses, with a maximum of k−1 biologically informative PC axes (Thia, 2023). All samples were plotted along the main discriminant functions (DFs) and examined visually. DAPC methods and results are reported according to recommended standards (Miller, Cullingham & Peery, 2020; Thia, 2023). R code for these analyses is included as Document S3.

Results

Sequencing of all libraries yielded 95.6 million reads with each library averaging 8.6 ± 2.9 million raw reads. After trimming and quality control, each library averaged 7.8 ± 2.4 million reads. MultiQC was used to assess sequence quality scores, GC and per-base sequence content, sequence length distributions, duplication levels, overrepresented sequences, and adapter content; all passing all acceptable threshold checks. A total of 168,921 SNPs were called between both the mitochondrial and nuclear data sets. 2,186 SNPs were multiallelic, and 616 were insertions and deletions (INDELs). INDELs and multiallelic SNPs remain a challenge for quantification, so this analysis is restricted to biallelic SNPs. Visualizations of FST values created by assessPool allow for the identification of outlier SNPs, but no significant outliers were present in the data set.

We used ezRAD because this approach provides a compromise between low coverage whole genome sequencing and typical reduced representation genomic approaches (RAD) that produce stacks of a handful of high coverage loci (Toonen et al., 2013), such that we can assemble long contigs, often yielding entire mitochondrial genomes (Forsman et al., 2017). We were able to recover a consensus of the entire mitochondrial genome for all pools to be analyzed separately from the nuclear dataset (sensu Forsman et al., 2017). Mitogenome comparisons yielded 276 SNP loci, but most of these SNPs did not meet our filtering criteria, leaving 23 high quality SNPs with a minimum threshold of 30x coverage. SNPs were validated by comparing reconstructed control region sequences from our pool-seq data with the control region sequences (GenBank # KM267565 –KM267626) for each individual reported in Clarke et al. (2015) as outlined in Kraft et al. (2020). Shared SNPs are the same between both studies, and limiting our analyses to those SNPS within the control region amplified by Clarke et al. (2015) produces the identical result reported in the previous study. Additionally, examining only the shared sampling locations, our results are highly correlated (Mantel test, r2 = 0.96, p < 0.05) with previous results (Clarke et al., 2015; Kraft et al., 2020), lending confidence to these results. After conservative filtering using the parameters listed in Document S2, a total of 23 SNPs were used for comparisons among sites in the mitogenome. From the nuclear data set, there were a total of 168,645 SNPs. Using the same conservative filtering parameters as for the mitochondrial genome (Document S2), 854 nuclear SNPs remained for allele frequency calculations; all retained SNPs were called in at least 90% of all pooled libraries at a minimum of 30X mean read depth (Table 1).

Table 1 Summary of pooled sequence data for post-filtered SNPs.

Includes number of individuals per pool (N), mean read depth per locus per pool, number of loci per pool, and number of SNPs per pool.

Pool name	N	Mean read depth (nDNA)	# Loci (nDNA)	# SNPs (nDNA)	
Red Sea (RDS)	50	271.6	277	847	
Indian Ocean (IDO)	70	127.9	283	854	
Taiwan (TAI)	47	165.6	283	854	
Papua New Guinea (PNG)	74	156.0	283	854	
North Central Pacific (NCP)	69	171.6	283	854	
South Central Pacific (SCP)	67	99.0	283	854	
Eastern Pacific (EPAC)	70	147.9	282	852	
Gulf of Mexico (GOM)	41	193.7	283	854	
Northwest Atlantic (NWA)	33	92.9	283	854	
Western Africa (AFR)	71	138.8	283	854	
Brazil (BRA)	36	144.2	283	854	
Total	628	1,709.2	283	854	

Figure 2 Pairwise FST values for all comparisons among silky shark (Carcharhinus falciformis) collection sites.

Upper left corner shows FST values from mitochondrial loci with a gradient from light blue being lower to dark blue being higher values. Lower right corner shows values estimated from nuclear loci with yellow being lower values to red being the higher values. Sample site abbreviations: RDS, Red Sea; IDO, Indian Ocean; TAI, Taiwan; PNG, Papua New Guinea; SCP, South Central Pacific; NCP, North Central Pacific; EPAC, Eastern Pacific; GOM, Gulf of Mexico; NWA, Northwest Atlantic; BRA, Brazil; AFR, Africa.

FST values from the mitochondrial data set were higher than those from the nuclear data set, sometimes by an order of magnitude (Fig. 2, Table 2). We also observed diagnostic (fixed) differences between Atlantic and Indo-Pacific populations, resulting in very high FST values (Table 2). Additionally, there was stronger mtDNA structure among populations within the Atlantic than within the Indo-Pacific (mean FST 0.094 ± 0.041 and 0.045 ± 0.022, respectively). Although nDNA structure showed the same pattern, the magnitude was less substantial, with a mean FST of 0.025 ± 0.006 among Atlantic comparisons and a mean FST of 0.022 ± 0.007 among Indo-Pacific comparisons. All comparisons between Atlantic and Indo-Pacific regions were higher than any comparison within an ocean basin (Table 2). The analysis of molecular variance (AMOVA) also demonstrated significant population structure among ocean basins (p = 0.01), RFMOs (p = 0.004), and the groups identified by DAPC (p = 0.01). However, the magnitude of variation explained by each AMOVA analysis differed, with 32.1% between ocean basins, compared to 67.9% within (Table 3). The least amount of variation was explained among RFMOs (22.2% compared to 77.8% of the observed variation within them), whereas the seven groups identified by DAPC analyses explained the greatest proportion of molecular variance at 34.8% among groups.

The Northwest Atlantic site showed the highest isolation based on both mtDNA and nDNA mean FST in comparison to other Atlantic sites, and the lowest mean mtDNA FST in comparison to any Indo-Pacific locations. Whereas most mitochondrial comparisons between the Atlantic and Indo-Pacific ranged from FST = 0.60–0.84, Northwest Atlantic samples ranged from FST = 0.38–0.46 (Table 2), with similar patterns in the nuclear data showing the Northwest Atlantic at intermediate values. However, the Northwest Atlantic also had the smallest sample size at 33 individuals (Fig. 1). The highest FST values between the Atlantic and Indo-Pacific were among comparisons including Brazil. The lowest comparison of Brazil to any Indo-Pacific region was higher than any other inter-ocean comparison except in the nuclear data set, where pairwise FST exceeded 0.06 between the Northwest Atlantic to Indian Ocean and the Northwest Atlantic to Eastern Pacific. Within the Indo-Pacific, the Indian Ocean and Eastern Pacific showed relatively higher isolation. Comparisons within the jurisdiction of the WCPFC (Western Central Pacific) yielded the lowest inter-ocean isolation in both data sets, yet all FST values among sampling locations were still significant. The Red Sea was more differentiated from the Indian Ocean (nDNA mean FST = 0.035, mtDNA mean FST = 0.062) and from the Eastern Pacific (nDNA mean FST = 0.031, mtDNA mean FST = 0.075) than any other Indo-Pacific regions (nDNA 0.015–0.02, mtDNA 0.02–0.04).

Table 2 FST value summary within and between ocean basin for Carcharhinus falciformis.

	mtDNA F ST values	nDNA F ST values	
	Range	mean (SD)	Range	mean (SD)	
Inter Atlantic	0.049–0.160	0.094 (0.041)	0.018–0.033	0.025 (0.006)	
Inter Indo-Pacific	0.012–0.082	0.045 (0.022)	0.014–0.035	0.022 (0.007)	
Between Atlantic and Indo Pacific	0.383–0.844	0.642 (0.144)	0.042–0.078	0.054 (0.009)	
All Comparisons	0.012–0.845	0.354 (0.312)	0.014–0.078	0.038 (0.018)	

Table 3 Analysis of Molecular Variance (AMOVA) of eleven pooled silky shark libraries, with respect to grouping by (a) ocean basin (Atlantic and Indo-Pacific), (b) RFMO, and (c) DAPC identified clusters.

Source of variation	Degrees of freedom	Sum of squares	Mean of squares	Variance component	% of variance	P-value	
(a)	
Between oceans	1	299.82	299.82	41.61	32.1	0.001	
Within oceans	9	792	88	88	67.9	
Total	10	1,091.82	109.18	129.61	100	
(b)	
Between RFMO	3	464.57	154.86	25.63	22.24	0.004	
Within RFMO	7	627.25	89.61	89.61	77.76	
Total	10	1,091.82	109.18	115.24	100	
(c)	
Between DAPC clusters	6	797.82	132.97	39.25	34.81	0.001	
Within oceans	4	294	73.5	73.5	65.19	
Total	10	1,091.82	109.18	112.75	100	

These inter- and intra-ocean-basin patterns are reflected in the principal component projection (Fig. 3), which show clear separation between the four Atlantic and the seven Indo-Pacific pools. However, samples within RFMOs do not consistently cluster together. For example, the Red Sea and Indian Ocean samples within the IOTC appear more divergent from one another than either is to any of the sampling locations throughout the WCPFC (Fig. 3). In the discriminant analysis of principal components (DAPC), the first eight PCs of the PCA were used and a single discriminant function was retained, yielding a proportion of conserved variance of 0.887 and assignment of 100% of pools. Seven genetic clusters were identified from our dataset using k-means clustering, with only samples in the WCPFC falling within the same cluster. Clusters within the Atlantic and Indo-Pacific were clearly distinct, being on opposite sides of the plot (Fig. 4). However, samples collected within the ICCAT and IOTC were also assigned to different clusters, as were each of the samples within the ICCAT.

Figure 3 Principle component analysis (PCA) plot of major allele frequencies for all pool-seq SNPs among samples collected within Regional Fisheries Management Organizations.

Sampling sites are color coded by RFMO as per the map (Fig. 1). Closely clustered pools of IOTC and WCPFC samples in top left include the Indian Ocean (IDO), Papua New Guinea (PNG), Taiwan (TAI), North Central Pacific (NCP), and South Central Pacific (SCP) sampling locations.

Figure 4 Discriminant analysis of principal components (DAPC) of silky shark sampling locations.

DAPC clustering for all pool-seq SNPs to identify genetic clusters. X-axis tick marks correspond to regionally pooled samples identified in Fig. 1, and color-coding corresponds to ocean-basin highlighting the distinctiveness of Atlantic and Indo-Pacific populations.

Discussion

Waples (1998) was the first to point out that genomic scale data has so much power that significant population genetic structure may not be biologically meaningful. Waples & Gaggiotti (2006) subsequently identified criteria for biological relevance of low, but significant, population genetic structure. First, for populations to be ecologically or demographically independent, the fraction of migrants (m) must be less than 10% of the total population size (Hastings, 1997). For many wildlife populations, that means the number of effective migrants (Nem) must be less than ∼1–25 individuals per generation (also see Mills & Allendorf, 1996; Wang, 2004). FST values reported herein, like those for most marine species, are relatively low. This leads to questions about biological as opposed to statistical significance and provides challenges for interpretation and management recommendations (Waples, 1998; Bird et al., 2011; Crandall et al., 2019). In its simplest form, FST should range between 0 and 1 and provide inference about the magnitude of gene flow among populations, but a multitude of studies have highlighted the complexities of estimating and interpreting FST values (Whitlock & McCauley, 1999; Hedrick, 2005; Jost, 2008; Holsinger & Weir, 2009; Meirmans & Hedrick, 2011; Bird et al., 2011). Numerous alternative measures of genetic differentiation or methods to rescale FST to simplify interpretation have been proposed, but all suffer from the same limitation that the maximum value that can be calculated scales to the frequency of the most frequent allele and is almost always well below one (Bird et al., 2011; Alcala & Rosenburg, 2019). Given such complexity in determining whether low FST values are biologically significant, many have turned to simulations to determine the scale of migration that would equate to observed FST values as a way to guide management recommendations. For example, in a multispecies study of population connectivity across the Hawaiian Archipelago, Crandall et al. (2019) used simulations to show that gene flow of ≤100 migrants per generation across the Hawaiian Archipelago resulted in an FST∼0.002. Likewise, Spies et al. (2018) performed simulations on populations of cod, and found FST values on the order of 0.0175 to 0.0008, using 50 to 500 migrants annually. These values are lower than we observe among populations of silky sharks within RFMOs here, but as outlined above, the number of migrants depends on both the population size and the migration rate (Nem). For example, Nem = 100 for both Ne = 10,000 & m = 0.01 and for Ne = 1,000,000 & m = 0.0001 (Crandall et al., 2019). Although we lack location-specific estimates of silky shark abundance, they are among the world’s most abundant shark species (Bonfil, 2008), and despite global decline (Rigby et al., 2021), the bycatch mass of silky sharks (Clarke et al., 2018; Lopez et al., 2020) implies that population sizes are well in excess of 10,000 individuals, constraining m to be less than 0.01. Thus, even within broad order of magnitude estimates, the population structure we find within RFMOs here is consistent with previous work showing that gene flow is limited, and these populations fall firmly within the realm of biologically meaningful and demographically independent populations (Hastings, 1997; Waples & Gaggiotti, 2006).

Significant FST values among every region sampled for this study indicate that silky sharks are not as dispersive as previously believed. Although we have only a single sample site from the IATTC, we find significant population structure within every other RFMO from which we have multiple samples, suggesting multiple distinct populations with limited exchange exist within each of the IOTC, WCPFC, and ICCAT management areas. Based on the criteria outlined above, these populations may comprise separate stocks which RFMOs could manage separately to avoid population declines. This is a clear case where a superficial understanding of life history led to the erroneous assumption of stock structure on the oceanic scale typical of pelagic fishes, and highlights the need for additional studies to map the boundaries of stock structure for this species. The polar opposite finding is available from the whitetip reef shark (Triaenodon obesus), which is sedentary and highly site-attached, yet maintains high connectivity across the archipelagos of the Central Pacific (Whitney et al., 2012). In both cases, a fragmentary understanding of life history would lead to ineffective management policies.

Population structure

Significant genetic population structure was found between every region sampled in the Atlantic. This includes the proximal geographical samples from the Gulf of Mexico and Northwest Atlantic. Two previous mtDNA studies, Clarke et al. (2015) and Domingues et al. (2017), both used the same individuals for population structure analysis in these regions. Neither study found any significant structure between the Gulf of Mexico and Northwest Atlantic. Domingues et al. (2017), however, did find population structure between the South Atlantic and North Atlantic using mtDNA control region data (and additional regions and sampling), but did not detect structure within southern or northern hemispheres. Here, we expand on the analysis of individuals previously published by Kraft et al. (2020) and add seven additional sampling locations to create a global survey of silky sharks using a pool-seq approach. Our results dramatically increase the amount of data available for this species across all these regions, and reveal finer-scale population structure than previously recognized. It is noteworthy that the results we present here are consistent with previous patterns of population genetic structure reported in Kraft et al. (2020) and Clarke et al. (2015), but the magnitude of structure differs due to more stringent filtering used herein to ensure SNPs were represented in each population. Despite variation in magnitude, the patterns and relative differences among populations observed in the data are consistent between studies (r2 = 0.96), as are the inferences and conservation recommendations drawn from them.

Similar to the results from the Atlantic regions, all sampled Indo-Pacific regions show significant FST values, which indicates population genetic structure within the Indo-Pacific despite the semi-pelagic lifestyle of this species. The FST values for regions sampled within the WCPFC domain were lowest, albeit still statistically significant. These regions showed stronger structure when compared to more distant samples from the Eastern Pacific and Indian Ocean. However, the Red Sea had lower FST values in comparison to the Western Pacific than to the adjacent Indian Ocean, indicating that forces other than simply distance act to structure these populations. Anomalously, Red Sea to Indian Ocean comparisons showed some of the highest FST values in the Indo-Pacific mtDNA data set except for the Eastern Pacific. This anomalous divergence runs counter to both the geography and the known biogeographic history of the Red Sea, which is thought to be recolonized from the Indian Ocean after desiccation events driven by glacial sea level change (DiBattista et al., 2013; DiBattista et al., 2016) (but also see Coleman et al., 2016). However, these counterintuitive results are also consistent with previous work by Clarke et al. (2015), which reported no significant difference between the Red Sea and Central Pacific (Line Islands) but did find significant differences between the Red Sea and Indian Ocean (Andaman Sea). It is unlikely this anomalous pattern results from higher connectivity between the Central Pacific and the Red Sea than with the geographically closer Indian Ocean and Coral Triangle. One possible alternative explanation is that intense fishing pressure in the Indian Ocean (Amandè et al., 2008) has accelerated allele frequency changes through genetic drift on greatly diminishing populations or through fisheries induced selection (see discussion in Conover & Munch, 2002;Kuparinen & Merilä, 2007; Rose et al., 2001).

Discriminant analysis of principal components (DAPC) clearly separates the ocean basins, which show 100% assignment of sites, highlighting how distinct these populations are from one another (Fig. 4). However, DAPC clustering also shows substantial differentiation among sampling sites within each of the IOTC and ICCAT (Fig. 3). Often used as a statistical method to analyze the genetic structure of populations and identify discrete groups through k-means clustering (Miller, Cullingham & Peery, 2020; Thia, 2023), DAPC finds support for seven populations among our 11 sampling locations throughout the 4 RFMOs. The magnitude of population structure (pairwise FST), the proportion of variation explained by the AMOVA, and divergence among sampling locations in the DAPC analyses within the same RFMO can exceed those measured between different RFMOs. For example, in the DAPC analysis, samples collected within the ICCAT show as much divergence from one another as all the samples from the IOTC, IATTC and WCPFC combined, whereas samples from IOTC cluster with the WCPFC (Fig. 3). De novo k-means clustering identified seven groups, with only samples from the WCPFC all falling within the same cluster. These seven groups also explain the greatest proportion of molecular variance in our AMOVA analyses (34.8%). However, the DAPC cluster that contains all the WCPFC samples also clusters with one sampling location from the IOTC. Thus, these genetic data fail to support any RFMO with more than a single sampling location as a single population. We recognize the limitations of our opportunistic sampling and pool-seq analysis which might miss important stepping-stone populations and precludes more sophisticated individual-based or clustering analyses, but these data emphasize that additional studies are urgently needed to understand how stocks of these pelagic predators are distributed among current fisheries management units.

Genetic separation between the Atlantic and Indo-Pacific is not surprising and consistent with all previous studies for C. falciformis. Likewise, using the same samples and limiting our analyses to the same SNPs, we get the identical results as both Clarke et al. (2015) and Kraft et al. (2020). However, our results reported here differ in magnitude because we apply more stringent filtering to ensure loci are shared among global samples. Many pelagic fishes have a primary population genetic partition at the Indian-Atlantic boundary (e.g., Graves & McDowell, 2015; Hirschfeld et al., 2021). The high FST values observed between oceans (ranging from 0.383–0.844 for mtDNA and 0.042–0.078 for nDNA) invokes the possibility of distinct evolutionary lineages in the Atlantic and Indo-Pacific. Though this question is outside the scope of our study, in addition to delineating fine-scale stock structure, it also seems worthwhile to investigate the evolutionary partitions in silky sharks between these two ocean basins.

Higher FST values with mtDNA than nDNA, as we report here for silky sharks, is a recurring theme in population genetics (Jorde, Bamshad & Rogers, 1998; Bird et al., 2011; Avise, 2012). Many of these results can be attributed to the effective population size in haploid mtDNA, which is four-fold lower than diploid nDNA (Hartl & Clark, 1997). In every generation, four nuclear alleles can be transmitted from diploid parents, whereas only one mtDNA allele can be transmitted from the maternal parent. The stronger genetic drift in mtDNA almost invariably yields higher FST values, making it the marker of choice for many studies of population genetic structure (Avise, 2012). Natural history can also play a role in highly structured mtDNA patterns. This mtDNA/nDNA pattern is apparent in several large migratory sharks, including the white shark (Carcharodon carcharias, Lamnidae; Pardini et al., 2001), shortfin mako (Isurus oxyrinchus, Lamnidae; Schrey & Heist, 2003), sandbar shark (Carcharhinus plumbeus, Carcharhinidae; Portnoy et al., 2010), and scalloped hammerhead shark (Sphyrna lewini, Sphyrnidae; Daly-Engel et al., 2012). In these species, higher mitochondrial structure is attributed to female site fidelity to reproductive habitat. Females and males may have similar migratory patterns, but only the males disperse gametes during these migrations through opportunistic mating (e.g., Bowen et al., 2005). Hence, the findings presented here invoke the possibility of female natal site fidelity and male mediated gene dispersal in silky sharks. Additional evidence for possible female site fidelity and multiple silky shark stocks per RFMO is also evident in life history data (Grant et al., 2019). To test the theory of female natal site fidelity and further understand genetic population structure of silky sharks, young of the year sharks captured in or near nursery habitats should be genetically surveyed as they would not have the time nor mobility required to travel long distances and intermix with sharks from other natal sites. Similar young of the year analyses have been performed on bluefin tuna to delineate Gulf of Mexico and Mediterranean population structure (Puncher et al., 2018), and sea turtles (Bowen et al., 2005). Tag/recapture studies would also be valuable to test for migration among RFMOs or the possibility of gender differences in dispersal.

Life history

Our genomic results are supported by other lines of evidence to suggest multiple populations within current fisheries management units. For cosmopolitan shark species, differences in life history parameter estimates between regions can provide insight into population structure and can be used to endorse regional management within or between ocean basins (Lombardi-Carlson et al., 2003; Smart et al., 2015). Grant et al. (2019) completed an intraspecific demographic analysis of silky sharks using life history parameter estimates available from several regions overlapping with the present study, including the Gulf of Mexico (Branstetter, 1987; Bonfil, Mena & De-Anda, 1993), Taiwan (Joung et al., 2008), Eastern Pacific (Sánchez-de Ita et al., 2011), Central Pacific (Oshitani, Nakano & Tanaka, 2003), Papua New Guinea (Grant et al., 2018), and the Indian Ocean (Indonesia Hall et al., 2012). These studies demonstrate that life history parameters, and subsequent demographic attributes (e.g., intrinsic population growth, generation time etc.), vary throughout the silky shark range, indicating demographic isolation and distinct stocks, consistent with the magnitude of population genetic structure observed in the present study. Although differences in methodologies across studies could account for some of the differences in population parameters (Grant et al., 2019), our genetic results seem to corroborate those findings such that population structure could be driving natural variation in life history parameters. Location-specific differences in life history parameters within the Western and Central Pacific suggest discrete populations perhaps due to natural variation or historical exposure to varied levels of fishing pressure. Differences in silky shark life history throughout their range are corroborated by observed genetic distinctions among locations, and together support the hypothesis that there are multiple stocks per RFMO jurisdiction, especially within the WCPFC.

Silky shark movement and habitat use

Habitat use and movement data for highly mobile or migratory species are commonly used for determining or refining the boundaries among stocks for management (Daly-Engel et al., 2012; Hilborn, 2012; Hays et al., 2019). For example, blue sharks (Prionace glauca, Carcharhinidae) and whale sharks (Rhincodon typus, Rhincodontidae), true roamers of the sea, have long distance movements documented by satellite tags: 28,000 km and 20,000 km respectively (Eckert & Stewart, 2001; Vandeperre et al., 2014; Vandeperre et al., 2016; Guzman et al., 2018). Concordant with these tracking studies, genetic surveys of both of these sharks reveal no population structure across the Indo-Pacific nor within the Atlantic Ocean (Castro et al., 2007; Schmidt et al., 2009; King et al., 2015; Taguchi et al., 2015; Yagishita, Ikeguchi & Matsumoto, 2020). Although tracking data for silky shark dispersal is limited, the three longest movements documented by satellite tags are 2,200 km, 3,195 km, and 4,755 km (Schaefer et al., 2019; Lara-Lizardi et al., 2020; Salinas-de León et al., 2024, respectively) which, if the norm, is inconsistent with the significant population structure we report here. However, these long distances may be exceptional when compared to most observations. Telemetry data up to 180 days typically reveal short term fidelity to drifting FADs and relatively short movements (under 1,000 km) away from tagging locations for silky sharks (Filmalter et al., 2011; Filmalter et al., 2015; Hutchinson et al., 2015; Hutchinson et al., 2019; Schaefer et al., 2019). Other studies using acoustic tags with fixed receivers on reefs or banks show most silky sharks demonstrated relatively long residence times and close association to tagging location (Clarke, Lea & Ormond, 2011; Hueter et al., 2018; Lara-Lizardi et al., 2020). Although additional longer-term satellite tracking data would help resolve migratory pathways if they occur, the majority of currently available tracking data indicates restricted movement patterns relative to truly pelagic sharks. The shorter movement patterns observed from silky sharks suggest smaller home ranges and therefore smaller isolated populations, congruent with differences in the life history characteristics and the genetic results reported in this study.

Management implications

Given that only a few effective migrants per generation are needed to homogenize genetic structure between neighboring populations (Mills & Allendorf, 1996; Hartl & Clark, 1997; Vucetich & Waite, 2000; Wang, 2004), the genetic structuring reported here indicates limited exchange among sites sampled within each of the RFMO management areas. Based on previous work reviewed above (Waples & Gaggiotti, 2006; Spies et al., 2018; Crandall et al., 2019), the magnitude of exchange among populations is unlikely to exceed a few percent of the population per generation and is consistent with observed life history variation. Thus, results of this study call into question the single stock per RFMO management scheme currently in place for silky sharks, instead providing evidence for multiple groups within each of the IOTC, WCPFC, and ICCAT jurisdictions. Further investigation into genetic population structuring within the IATTC region is warranted, given only a single collection site from within that RFMO. Management regimes should be matched to demographically independent populations to the greatest extent possible. Based on the collective evidence of population genomics, life history differences and mean dispersal of tagged individuals, available data indicate that depleted populations within an RFMO management region would most likely need to recover primarily via local recruitment rather than relying on immigration from neighboring locations, as might be expected for highly motile pelagic sharks. Given the long generation time of silky sharks (9.54–19.34 years; Grant et al., 2019), low migratory rates within this timeframe would not sustain nor replenish populations under current exploitation levels in these RFMO jurisdictions. DAPC analyses support at least seven populations among our 11 regional sampling locations, which explains significantly more of the observed population genetic structure in the AMOVA analyses than is explained by RFMOs. We find significant population genetic structure within the jurisdiction of every RFMO from which we have more than a single sampling site. Overall, genetic results presented here, reinforce available life history and movement data that silky sharks show much greater site fidelity than other pelagic sharks and are less dispersive than previously believed. Thus, our data highlight discordance between observed population genetic structure and management regimes, highlighting the need for a detailed study to accurately identify stock boundaries.

Conclusion

Accurate estimation of stock size and exploitation rates are critical to sustainable fisheries management (Hilborn, 2012), as is an understanding of migration among stocks (Hastings, 1997). This study finds fine-scale population genetic structure within each of the current management boundaries from which there is more than a single sample. Notably the populations identified here are based on specimens clustered by regional biogeographic provinces, with DAPC analyses supporting at least seven groups among our 11 regional samples. The fact that every sample location included in this study shows significant population structure from the others almost certainly indicates that finer scale studies would likely reveal additional population units. We did not have the resources to survey northern and southern hemispheres in the Eastern Pacific Ocean which, given the patterns observed in other marine species are likely to constitute distinct stocks, and should be a priority for future study. Our population genetic data are consistent with previous life history parameters and telemetry data, none of which support the conclusion that there is a single stock of silky sharks per RFMO, so we advocate for additional study to define the stock boundaries for silky sharks. A revised fisheries management approach based on accurate delineation of silky shark stocks within these RFMO jurisdictions could assist in rebuilding depleted silky shark populations.

Supplemental Information

Supplemental Information 1 Individual Sample Metadata

Supplemental Information 2 SNP Filtering Parameters

Supplemental Information 3 Silky Shark Analysis R Code

This study was made possible by extensive sample collections initiated by authors; JDF, MH, JLYS, JRH, SG and the generous donation of specimens by C.R. Clarke, M. Shivji, S.A. Karl, J. Martinez and through collection efforts by the Pacific Islands Regional Observer Program, IATTC Observer Program, American Samoa Regional Observer Program, and Southeast Shark Bottom Longline Observer Program. Samples were also collected by MH and JDF in the IDO, SCP and AFR during chartered research cruises aboard commercial fishing vessels supported by the International Seafood Sustainability Foundation. We thank members of the ToBo Lab for sharing expertise, advice, and discussions that improved this manuscript. We also thank K. Freel for providing edits to the manuscript to ensure clarity. We thank the staff of the HIMB EPSCoR Evolutionary Genetics Core Facility and especially A. Eggers and M. Mizobe for assistance with genotyping. Special thanks to D. Lerner, K. Holland, C. Meyer, D. Bethea, D. McCauley, and C. Wilson for guidance and sage advice. Thanks to the staff at Hawaiʻi Institute of Marine Biology for their support throughout this project. This is contribution #1944 from the Hawaiʻi Institute of Marine Biology and contribution #11767 from the School of Ocean and Earth Science and Technology at the University of Hawaiʻi at M­anoa, and UNIHI-SEAGRANT-4861 from the University of Hawaiʻi Sea Grant Program.

Additional Information and Declarations

Competing Interests

Author Contributions

Animal Ethics

DNA Deposition

Data Availability

Robert J. Toonen is an Academic Editor for PeerJ. Simon Gulak is employed by Riverside Technology, Inc and John R. Hyde is employed by National Marine Fisheries Service.

Derek W. Kraft conceived and designed the experiments, performed the experiments, analyzed the data, prepared figures and/or tables, authored or reviewed drafts of the article, and approved the final draft.

Emily E. Conklin conceived and designed the experiments, performed the experiments, analyzed the data, prepared figures and/or tables, authored or reviewed drafts of the article, and approved the final draft.

Evan B. Freel conceived and designed the experiments, performed the experiments, analyzed the data, prepared figures and/or tables, authored or reviewed drafts of the article, and approved the final draft.

Melanie Hutchinson analyzed the data, authored or reviewed drafts of the article, and approved the final draft.

Julia L.Y. Spaet analyzed the data, authored or reviewed drafts of the article, and approved the final draft.

Robert J. Toonen conceived and designed the experiments, analyzed the data, prepared figures and/or tables, authored or reviewed drafts of the article, and approved the final draft.

Zac H. Forsman conceived and designed the experiments, analyzed the data, prepared figures and/or tables, authored or reviewed drafts of the article, and approved the final draft.

Michael I. Grant analyzed the data, authored or reviewed drafts of the article, and approved the final draft.

John David Filmalter analyzed the data, authored or reviewed drafts of the article, field collections, provided expertise in shark natural history, and approved the final draft.

John R. Hyde analyzed the data, authored or reviewed drafts of the article, field collections, provided expertise in shark natural history, and approved the final draft.

Simon J.B. Gulak analyzed the data, authored or reviewed drafts of the article, field collections, provided expertise in shark natural history, and approved the final draft.

Brian W. Bowen conceived and designed the experiments, analyzed the data, authored or reviewed drafts of the article, and approved the final draft.

The following information was supplied relating to ethical approvals (i.e., approving body and any reference numbers):

Samples were opportunistically collected aboard commercial fishing vessels or from fish markets.

The following information was supplied regarding the deposition of DNA sequences:

The sequences are available at NCBI: SRR25394102–SRR25394112, PRJNA997384.

The following information was supplied regarding data availability:

The sequences are available at NCBI: SRR25394102–SRR25394112, PRJNA997384.

The assessPool code is available at Github and Zenodo:

- github.com/ToBoDev/assessPool

- Freel, E., & Conklin, E. (2024). assessPool. Zenodo. https://doi.org/10.5281/zenodo.13363639.

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
