# Peer review of "Global stock structure of the Silky shark (Carcharhinus falciformis, Carcharhinidae) assessed with high-throughput DNA sequencing"

_PeerJ, doi:10.7717/peerj.19493_

## Round 0.1 · original submission · Major Revisions

· Academic Editor

Major Revisions

Your manuscript has been reviewed and as you noticed reviewers highlighted points in need to be clarified. For instance, a further explanation of the methodology is needed, the number of samples is crucial to justify the study, and clarify the issue of "reference genome". A major concern is how these new data analysed are not overlapped with a previous publication (Kraft et al. 2020). Consequently, authors need to add more data to the work. Conclusions need to be less ambitious and in proportion to the data.

Please, respond in detail to all corrections and inquiries provided by the reviewers, and if you feel in a position to further respond to each of these observations you can submit your work for another round of revision.

Reviewer 1 ·

Basic reporting

No comment.

Experimental design

I believe there are issues with the genetic data generated (please see the validity of the findings section) and some methods are not described in sufficient detail.

There is insufficient data on where the samples were collected (e.g., coastal or offshore locations, including potential nurseries and/or aggregations sites with kin) and what life history stages the animals were sampled at. The latter is crucial to studies of population genetics in highly mobile species such as silkies and providing these data would allow for further assessment of the levels of genetic structure.

The authors mentioned using a “reference genome” that the reads were mapped to, but there is no description of how this reference was assembled. In particular, it’s unclear which samples were used to assemble it. Was it a subsample of samples from all RFMOs? Also, to avoid confusion with an actual reference genome, I would refer to this as a “genomic reference”.

In addition to PCA, I suggest running discriminant analysis of principal components with k-means clustering to group the samples.

Please describe how assessPool allows for the identification of outlier SNPs. I recommend running additional methods to assess for FST outliers, if possible.

Validity of the findings

While I concur with the authors that there are likely to be multiple genetic stocks within one or more of the RFMOs, I believe there are severe issues with the genetic data they have generated. For both the mitochondrial and nuclear data, the estimates of FST between regional locations are inconsistent with other population genetics studies of sharks, including silkies. Therefore, I am concerned that the observation of significant structure between each location is due to inaccurate estimates of allele frequencies rather than reproductively isolated units warranting separate management.

The time and resources available to managers are limited, so it is vital that studies informing changes in management are robust. If not, attention may be unduly focused on some populations at the expense of others. Moreover, the excessive identification of genetic stocks can reduce confidence in assessments of genetic structure and their usefulness for managers.

I expand upon some of the issues below:

1. The FST values reported are considerably higher than what one would expect for neutral population structure of sharks like silkies. In the western North Atlantic, for example, Portnoy et al. 2015 observed an FST value of 0.021 for neutral nuclear SNPs based on samples of bonnethead sharks from the eastern Gulf of Mexico and North Carolina. For the mitochondrial control region, ΦST was 0.064-0.161. Another study (Díaz-Jaimes et al. 2020) that used a distinct RAD protocol to Portnoy et al. found similar estimates of nuclear FST in bonnetheads between these regions. Given the differences in life histories and movement patterns between silkies and bonnetheads (see Kohler et al. 2019), your estimates of FST do not seem realistic. It’s possible that your nuclear data contain loci under selection, but you did not adequately assess for FST outliers.

2. Furthermore, the mitochondrial data is inconsistent with previous studies that used the exact same samples. Clarke et al. 2015 observed FST of 0.001 between the Atlantic and Gulf of Mexico based on 34 polymorphic sites of the mitochondrial control region. Your FST value is >100x greater than that and is based on fewer informative sites (23 SNPs). The manuscript does not describe how genetic structure was assessed using the mitochondrial data and I do not understand how you observed structure that is two orders of magnitude greater with fewer informative sites. Domingues et al. 2018 observed a similar level of differentiation between the Atlantic and Gulf to Clarke et al. 2015, and both studies suggest no structure between these regions.

3. Is it possible to reconstruct mitochondrial control region sequence data from the MiSeq reads to produce a dataset that is comparable to Clarke and Domingues? I’m unfamiliar with how ezRAD works, but if you could redo the analyses of Clarke and Domingues and show that there are similar levels of structure at the control region, I would have more confidence that the higher level of structure you observed is due to diversity outside of the control region, rather than an issue with library preparation and/or bioinformatics.

Additional comments

Raw data has been made available but analysis code has not. This is important for reproducible research.

Reviewer 2 ·

Basic reporting

The manuscript by Kraft et al. extends the dataset published by Kraft et al. 2020, PeerJ. In this new work, the authors have added 7 sampling points to the original dataset, but it is not clear if the four sites shared with the 2020 paper are made of newly sampled individuals or they are overlapping with previous work. I find the manuscript important as it is urgent to add more data on different shark species with different life history traits to improve our understanding of the influence they have on connectivity, which in turns is of major interest when it comes to devise coherent managing program. However, I found the conclusions disproportionately strong with respect of the evidence provided. The whole idea behind the paper is that since nuclear Fst are significant between all comparisons, it means that the studied regions need to be treated as independent populations deserving protection from overexploitation

Experimental design

I have three major issues concerning their conclusion:
- First, it is not clear how significance of Fst was assessed, but it is possibly my fault. The authors claim that they used a permutation test as in the original Excoffier et al. 1992 paper. In the classic AMOVA approach, individuals are randomly assigned to one of the two population (in a pairwise Fst), hence creating a null distribution of the Fst under panmixia. However, with the pool seq you do not have individuals, but only allele frequency, so I do not understand what the authors are permuting. I think it is very important to clarify this point. I would rather imagine (but of course this is just a guess) that the authors are rather bootstrapping loci. If this is the case (there are some R libraries typically following this approach, such as STAMPP, but I do not know what ade4 does), the authors are at best computing a confidence interval around their estimate, but they are not testing the hypothesis of panmixia. Since all their conclusion are based on the significance of the Fst, it is really important to explain what it has been done.
- Second, I compared the Fst values with those of the 2020 paper. I am not sure they are the same individuals, but the sample is almost of the same dimension and the molecular protocol is the same. For some comparison, there is almost a 2X difference, which suggest that the results are not stable. Indeed, a difference of 2X obtained by resampling could completely change the whole interpretation of this manuscript. In my opinion (ie., not having looked at the data), this suggest that the variance associated with the Fst computed from poolseq might be higher than suspected by the authors, in which case some care should be taken in interpreting the results (and the variance should be correctly computed).
- Third, the authors claim to have sequenced the whole mtDNA (as in the 2020 paper). I think it would be important to explain how, I really do not understand how with a Rad approach would be possible to obtain the whole mitochondrial genome. Once again, the comparison between the values here presented and the 2020 paper show, at minimum, that resampling individuals may significantly change the results. In other words, it is possible that the variance in the Fst estimate has not been correctly taken into account (the issue of significance applies also to mtDNA data, as they have been obtained by poolseq). More worrisome, the same individuals Sanger sequenced by Clarke et al. 2015 have large difference in the computed Fst. This could suggest an issue with the poolseq (which would then also apply to the nuclear dataset).

Validity of the findings

no comment

Additional comments

For all these reasons, I think that the results presented are not robust and the authors should better explain either the molecular and the statistical approaches adopted. I have also some minor comments:
- Line 165. Not clear which reference genome. If it exists a reference genome for the silky shark, please provide the entry (I could not find it). If not, explain what does this mean.
- Lines 211-212. Please remove this sentence, there is no reason to assume SNP are neutral because they are scattered through the genome.
- Line 214. Please explain how the whole mtDNA genome was assembled from the ezRAD
- Line 288. IBD could be tested, which I think would be a great idea.
- Lines 299-302. Please remove, this is too speculative. Selection does not act on the whole nDNA and there is no reason why drift should determine the observed pattern

---

## Round 0.2 · Major Revisions

· Academic Editor

Major Revisions

Your manuscript has been re-reviewed by the same referees and they still find many issues precluding it for acceptance. Please, you need to address in detail all suggestions raised by the referees and if you decide you can re-submit again. You need to consider that I will need to request another review round for having a final decision.

Reviewer 1 ·

Basic reporting

I disagree with the assertation that the vast majority of studies do not provide metadata associated with samples, such as time/date/location of collection and life-history stage, and believe these data are vital to fully evaluate the patterns of population structure reported here. Even though the individual-based metadata cannot be directly linked to individual samples within each pool, these data are still important to include, particularly for highly mobile species with ontogenetic habitat shifts. For example, if the majority of samples within each pool were collected from young individuals in coastal or aggregation locations likely to be used in early life stages, each pool would likely represent a reproductive unit and may contain kin. On the other hand, if the majority of samples were taken from reproductively mature individuals, it is possible that they were sampled outside of their stock of origin. In this situation, the pool would represent more of a mixed stock than a reproductive unit. Please see further discussion of this point below.

Experimental design

No comment

Validity of the findings

I have read Kraft et al. (2020) and endeavored to understand how they validated SNPs against the mitochondrial data produced by Clarke et al. (2015). My understanding is that Kraft et al. (2020) observed levels of mtDNA structure between the Red Sea and other locations (North Atlantic, Gulf of Mexico, and Brazil) that are similar to what Clarke et al. observed, but observed levels of structure 4-66 times greater among the other locations (e.g., GM:NA, 0.066 vs. 0.001). While the relative increase among these levels of structure may be consistent, the significantly elevated values could be due to bias that consistently skews comparisons not involving the Red Sea. Therefore, I’m concerned that the data presented in Kraft et al. (2020) and the current manuscript support conclusions that are in contrast to previous studies (e.g., Clarke et al. 2015), and have little reason to believe these studies have more robust data than Clarke et al.

Furthermore, it is not clear how structure was assessed for the mitochondrial dataset (the other reviewer asked for clarification about this too). From the revised changes I see in the manuscript (lines 207-221), the method used to assess mtDNA structure is not stated explicitly, but from what I understand, it appears that each mtDNA SNP was treated as an individual locus. Is this the case? My understanding is that mtDNA structure cannot be assessed this way because, unlike nuclear SNPs, each mtDNA SNP comes from the same locus.

If the authors can detail sampling information (e.g., basic descriptions of the habitats in which samples were collected, numbers of each life history stage sampled, and if any samples were collected in the same location on the same day) and describe in more detail an appropriate method that was used to assess mtDNA structure, I would be happy to review the manuscript again. However, I remain concerned with the validity of the results and conclusions that each sample constitutions a distinct genetic stock.

My concerns can be better understood by reviewing silky shark tagging data documented in Kohler et al. 2019 Marine Fisheries Review (page 78). Figure 37a shows multiple long-distance movements between the North Atlantic and Gulf of Mexico among 65 animals tagged. This indicates that the North Atlantic and Gulf constitute a mixed stock, and thus your pooled samples for these locations could contain individuals from the other location, which would remove any signal of genetic structure. This is likely to be the case if adult animals were sampled in pelagic habitats. By contrast, if pooled samples are predominantly composed of younger individuals that have not left their region of origin, I could foresee how this sampling scheme leads to the Gulf and Atlantic being observed as distinct stocks. In this situation, it would be helpful to know if the pools potentially contain related individuals (e.g., full and half siblings), because these would elevate estimates of structure and could explain why your estimates are much higher than expected. Siblings are more likely to be sampled in potential nursery habitats within a short period of time (e.g., same day). This is why sampling information is critical to properly evaluate patterns of genetic structure in sharks.

Reviewer 2 ·

Basic reporting

I re-reviewed the manuscript from Kraft et al. I have still the same concerns I had when reading the previous version. I still think that some clarity is lacking in some parts of the method section and I am still convinced that the results are overinterpreted. The authors claim from the Abstract that they report evidence for isolated population. I do not see how they show that these populations are isolated. The Fst remains low (0.078 being the largest pairwise comparison), which means that there is a (slight) differentiation in allele frequency, not that the populations are isolated. To prove that they are isolated, the authors should test demographic models in which migration is not ongoing anymore. Given the low Fst, I am pretty sure this is not the case, but without testing it is impossible to know. This is an important misunderstanding of Fst and its significance. I think the authors should limit their conclusions to the description of their data, without making evolutionary interpretation as no hypothesis was tested. Of course, this is my personal feeling, but I feel dangerous to spread the idea that based on Fst we could make such strong assumption about the history of a species.
I am also still not convinced by their Fst calculations: first, to reply to the rebuttal letter, Fst is based on allele frequency. AMova was introduced to take into account molecular distance, otherwise it is a classic frequency based estimation of Fst (but in a nested framework). Excoffier et al. proposed the permutation of individuals among population to test for the Fst significance: in their pooled sample used here, no individuals are present, so once again, the authors are not for sure following the Excoffier et al protocol. It would be nice if the authors could explain what they are permuting (not sure what “permuting allele frequency table” means).
It is also important that the authors provide the parameter used for the Rad assembly and that they state more clearly that they have not assembled a reference genome, but a set of Rad loci, which is not exactly the same thing. To make the analyses repeatable, assembly parameters and the whole pipeline should be explained. I also still do not understand why the whole mtDNA genome is retrieved and not the whole nuclear genome (almost) if the restriction enzyme is a frequent cutter. I would then expect that the nuclear loci retrieved do not have necessarily the same length, but no information is reported. At the same time, I guess that only part of the mtDNA is covered, and statistics about mapping and % of genomic coverage should also be reported.
Finally, I (of course) understand that using different filters may give different results. But if the global conclusion depends on the filter used, than there is a problem, since in general it is not easy to judge what the best filtering is. More worrisome, it is easy to compare the mtDNA results presented here to those of Clarke, where the data were produced by Sanger sequencing (so to say, the gold standard). For example, the authors computed an Fst of 0.16 between BRA and NWA, while Clarke et al. only of 0.031. This suggests (at best) that the filters used here are not optimal.
In synthesis, I think that some methods and results are still not well explained and some uncertainties remain on the results when compared to previous work. Also, once again, the results are wrongly interpreted as there is no formal test of isolation between populations, which are most likely still largely exchanging migrants. I do not think that the idea of isolation based simply on a Fst value (on top of that, quite small) should be conveyed in the scientific literature and I warmly encourage the authors to reconsider their interpretations.

Experimental design

no comment

Validity of the findings

no comment

---

## Round 0.3 · Major Revisions

· Academic Editor

Major Revisions

In light of your Appeal, we sought additional reviewers.

Please, find the comments of two new reviewers (and a previous reviewer, reviewer 1, who recommended rejection) about your manuscript. Respond to them accordingly and resubmit. Thank you.

· Appeal

Appeal


· · Academic Editor

Reject

Given the multiple opportunities provided for the authors to fulfill recommendations, a reviewer rises major concerns because of limited improvements to the manuscript. I am afraid I have to reject the manuscript in the present form.

Reviewer 1 ·

Basic reporting

This is the third time I have reviewed this manuscript and there have been limited improvements in the clarity of methods and conclusions. In the response to reviewers, the authors claim to have softened their language in terms of management recommendations, but there are multiple instances in which my reading leads me to think the opposite. In response to one particular comment, my objection is not only to the magnitude of specific differences but the overall patterns of population structure. Fundamentally, given the description of methods to generate and analyze the data, I am not convinced that the manuscript provides new and robust information on silky shark population structure.

Experimental design

Nothing to add.

Validity of the findings

Nothing to add.

Additional comments

Nothing to add.

Reviewer 3 ·

Basic reporting

The topic of silky shark population genetic structure is an important one given the high exploitation pressure on this species worldwide and corresponding need for improved management measures. Kraft and colleagues have provided the most geographically widespread assessment of population structure in this species, and their use of genomics scale markers from both nuclear and mitochondrial DNA provides the highest resolution assessment to date.
I reviewed the manuscript version containing author track changes addressing what appears to be a 2nd round of original reviewer comments. This version of the manuscript is well written, the Results are clearly presented, and the Discussion points are now well supported by the Results. The authors have adequately addressed the 2nd round of reviewer suggestions, which have resulted in an improved manuscript that I found informative.

Experimental design

The study is well designed. Obtaining samples from marine species that have globally widespread distributions is a well-known challenge, and the authors are commended for obtaining their broad geographic sample representation. The addition of the sample metadata provides a useful resource for interpretation of future studies that may be conducted on silky sharks.

Validity of the findings

With the latest round of revisions provided, including softening of their language claiming population isolation, clarifies their inferences. I also appreciated their additions on biological vs. statistical significance of their results.

Additional comments

My only suggestions for the author’s consideration are:

1. The word “resolved” in the manuscript's title following “Global” seems a little too definitive because there are regions where silky sharks occur that were not assessed – understandably given the difficulties of obtaining samples. I suggest considering a title that removes the word “resolved”. Perhaps something approximately along the lines of:
“Globally widespread assessment of the Silky shark (Carcharhinus falciformis, Carcharhinidae) with high-throughput DNA sequencing demonstrates a high level of population structure”

2. There are recent publications on silky shark genetics and migrations that are relevant to this study and could be included:

- Genetic stock structure of the silky shark Carcharhinus falciformis in the Indo-Pacific Ocean. PLoS ONE 18(10): e0292743. https://doi.org/10.1371/journal.pone.0292743

- Longest recorded migration of a silky shark (Carcharhinus falciformis) reveals extensive use of international waters of the Tropical Eastern Pacific. J Fish Biol. 2024; 105:378–3


3. Is there a citation for the following sentence?: "According to data from Hong Kong's Census and Statistics Department, 83 countries or territories supplied more than 10.3 million kilograms of shark fin products to Hong Kong in 2011. (ms lines 65-67).

4. Line 72: According the the Reference section, Beerkircher 2003 should be Beerkircher et al. 2003.

Reviewer 4 ·

Basic reporting

No comment

Experimental design

Based on lines 85 to 99 I understood the main question to be answered as "is there population structure between regions within the RMFOs". This could have maybe been summarised at the end of the paragraph to make it more clear.
I feel that additional analysis such as a description of private alleles between the populations or isolation by distance (not sure if possible with pooled SNPs) would complement the investigation which very much leans on FST values and one PCA. I did not see defined which p-value was considered significant for FST. As a DAPC is a supervised approach giving more weight to interpopulation than intrapopulation differences, it might be good to refer to following paper and clarify how the method works: Thia JA (2022) Guidelines for standardizing the application of discriminant analysis of principal components to genotype data.

Validity of the findings

The authors have produced really interesting insights into the global and regional population structure of Carcharhinus falciformis, and I think that this can be a very important part of evidence used for stock management. That said, I do think their conclusions are a little too bold based on the type and amount of analysis done. Pooling individuals together from opportunistic sampling (which I fully understand is for species like this often the only way to get samples) unfortunately means that we are missing out on some areas that might serve as stepping stones, or the possibility to do cluster analysis. I don’t think the presented data is enough to ask for a change in management. The interpretation of FST values is difficult as with increasing number of loci, fine-scale population structure becomes detectable that shows limitations in gene flow, but might not mean there are too few migrants to replenish a population. Here I think additional analysis is needed to demonstrate a lack of migration and recruitment between populations. The authors are writing themselves in lines 402 to 404 that sex-biased dispersal could be possible for the species, hence there is a limitation to what their data can tell us about migrants at this moment (as it unfortunately misses some meta data to inform on life stage and sex of each population). I was also surprised that the results of the DAPC analysis seem to be not discussed in the context of intraregional differences. Overall, there is too little analysis and too many limitations (which are also not explained enough) to draw the conclusions that there are “multiple genetic populations” (a term which is not really explained), I do however think that they can absolutely make a case that the structure they found should be considered and that further investigation into recruitment between populations is urgently needed to understand the stock management better.

---

## Round 0.4 · Major Revisions

· Academic Editor

Major Revisions

Thank you for your patience as I reviewed all of the versions of this manuscript in order to make a decision. After a careful evaluation, I have decided to give you one more opportunity to address the previous rounds of comments. In particular, please note that different reviewers have pointed out that the methods are not as clear as they could be and there is over-interpretation of your results, all of which I agree with. In your rebuttal, clearly show where you have attended to the reviewers' suggestions and concerns.

---

## Round 0.5 · accepted · Accept

· Academic Editor

Accept

I have read through your rebuttal letter and the resubmitted manuscript. I am satisfied with the changes and modifications that have been made to the manuscript. I especialy appreciate that you have taken time to clarify the methods and in particular softened the interpretation of the results. I am recommending that this manuscript be accepted for publication. Congratulations.